# Efficacy and tolerability of bevacizumab in patients with severe Covid-19

Jiaojiao Pang [1,15], Feng Xu [1,2,3,15], Gianmarco Aondio[4,15], Yu Li [5], Alberto Fumagalli [4], Ming Lu [2], Giuseppe Valmadre[4], Jie Wei[6], Yuan Bian [1], Margherita Canesi[7], Giovanni Damiani [8], Yuan Zhang [2], Dexin Yu [9], Jun Chen [10], Xiang Ji[5], Wenhai Sui [3], Bailu Wang [2], Shuo Wu [1], Attila Kovacs[11], Miriam Revera[12], Hao Wang [13], Xu Jing[1], Ying Zhang [1], Yuguo Chen [1,2,3✉] & Yihai Cao [14✉]

On the basis of Covid-19-induced pulmonary pathological and vascular changes, we hypothesize that the anti-vascular endothelial growth factor (VEGF) drug bevacizumab might be beneficial for treating Covid-19 patients. From Feb 15 to April 5, 2020, we conducted a single-arm trial (NCT04275414) and recruited 26 patients from 2-centers (China and Italy) with severe Covid-19, with respiratory rate ≥30 times/min, oxygen saturation ≤93% with ambient air, or partial arterial oxygen pressure to fraction of inspiration $O_2$ ratio ($PaO_2/FiO_2$) >100 mmHg and ≤300 mmHg, and diffuse pneumonia confirmed by chest imaging. Followed up for 28 days. Among these, bevacizumab plus standard care markedly improves the $PaO_2/FiO_2$ ratios at days 1 and 7. By day 28, 24 (92%) patients show improvement in oxygen-support status, 17 (65%) patients are discharged, and none show worsen oxygen-support status nor die. Significant reduction of lesion areas/ratios are shown in chest computed tomography (CT) or X-ray within 7 days. Of 14 patients with fever, body temperature normalizes within 72 h in 13 (93%) patients. Relative to comparable controls, bevacizumab shows clinical efficacy by improving oxygenation and shortening oxygen-support duration. Our findings suggest bevacizumab plus standard care is highly beneficial for patients with severe Covid-19. Randomized controlled trial is warranted.

[1] Department of Emergency Medicine, Shandong Provincial Clinical Research Center for Emergency and Critical Care Medicine, Institute of Emergency and Critical Care Medicine of Shandong University, Qilu Hospital of Shandong University, Jinan, Shandong, China. [2] Clinical Research Center of Shandong University, Jinan, Shandong, China. [3] Key Laboratory of Cardiovascular Remodeling and Function Research, Chinese Ministry of Education, Chinese National Health Commission and Chinese Academy of Medical Sciences, The State and Shandong Province Joint Key Laboratory of Translational Cardiovascular Medicine, Jinan, Shandong, China. [4] Department of Medicine and Oncology, Moriggia-Pelascini Hospital, Gravedona ed Uniti, Gravedona, Italy. [5] Department of Pulmonary and Critical Care Medicine, Qilu Hospital of Shandong University, Jinan, Shandong, China. [6] Department of Emergency Medicine, Renmin Hospital of Wuhan University, Wuhan, Hubei, China. [7] Department of Neurological Rehabilitation, Moriggia-Pelascini Hospital, Gravedona ed Uniti, Gravedona, Italy. [8] Department of Radiology, Moriggia-Pelascini Hospital, Gravedona ed Uniti, Gravedona, Italy. [9] Department of Radiology, Qilu Hospital of Shandong University, Jinan, Shandong, China. [10] Department of Radiology, Renmin Hospital of Wuhan University, Wuhan, Hubei, China. [11] Department of Intensive Care Unit, Moriggia-Pelascini Hospital, Gravedona ed Uniti, Gravedona 22015, Italy. [12] Department of Cardiology, Moriggia-Pelascini Hospital, Gravedona ed Uniti, Gravedona 22015, Italy. [13] Department of Critical Care Medicine, Qilu Hospital of Shandong University, Jinan, Shandong 250012, China. [14] Department of Microbiology, Tumor and Cell Biology, Karolinska Institutet, Stockholm 17177, Sweden. [15]These authors contributed equally: Jiaojiao Pang, Feng Xu, Gianmarco Aondio. ✉email: chen919085@sdu.edu.cn; yihai.cao@ki.se

Coronavirus disease 2019 (Covid-19) is an ongoing worldwide pandemic[1]. As of July 7, 2020, there were 11,468,979 confirmed cases with 535,181 deaths across 216 countries. Among the confirmed cases, 19%–30% were classified as severe[2,3]. Dyspnoea caused by inflammatory pulmonary effusion or edema presents in almost all patients with severe Covid-19 and instigates pulmonary and systemic hypoxia[4–6]. Although many commendable efforts have been made[7–9], no drug with demonstrated clinical efficacy for severe Covid-19 is available. Approaches of oxygen-support including mechanical ventilation, non-invasive ventilation, high-flow oxygen, and low-flow oxygen become indispensable, which are closely associated with long hospital stay[2–4,10], causing a worldwide shortage of ventilators and other respiratory support medical supplies. The serious pandemic situation poses a global challenge for medical supplies and demands an urgent need for developing effective drugs[11].

We propose a novel therapeutic concept that blocking vascular endothelial growth factor (VEGF) for treating patients with severe Covid-19 patients. Acute respiratory distress syndrome (ARDS) and dyspnea create hypoxia in lung tissues and other organs. Hypoxia induces VEGF expression through activation of the Prolyl hydroxylases (PHD)-hypoxia-inducible factor (HIF)-1 pathway, which upregulates VEGF expression through transcription activation[12]. VEGF is a potent vascular permeability factor that induces vascular leakiness in Covid-19-infected lung tissues, resulting in plasma extravasation and pulmonary edema, which further increases tissue hypoxia[13,14]. Additionally, VEGF participates in lung inflammation[15]. Blocking VEGF and the VEGF receptor (VEGFR)-mediated signaling would improve oxygen perfusion and anti-inflammatory response and alleviate clinical symptoms in patients with severe Covid-19. Thus, we employed bevacizumab, a humanized anti-VEGF monoclonal antibody, for treating patients with severe Covid-19.

Supportive clinical evidence for our therapeutic hypothesis includes: (1) patients with severe Covid-19 suffer from severe hypoxia; (2) VEGF levels in patients with severe Covid-19 are markedly elevated[5]; (3) pulmonary edema frequently presents in Covid-19 patients[16,17]; (4) autopsy analysis of Covid-19 patients shows excessive extravasates in alveoli of the infected lungs[18,19]; (5) vascular disorganization and endothelial cell proliferation in the Covid-19-infected lung tissues[18,19], suggesting VEGF-induced vascular effects; (6) overreactive inflammatory response[20]; and (7) experimental animal models demonstrate that anti-VEGF therapy improves pulmonary edema[21,22]. In confirmatory with our hypothesis, recent studies have also demonstrated vascular dysfunction of the Covid-19-infected tissues[23,24].

Bevacizumab has been used in clinical oncotherapy since 2004, with considerable reliability and safety. Taken together, we designed this trial to investigate the clinical benefits of bevacizumab plus standard care for treating patients with severe Covid-19.

## Results

**Study participants**. Twenty-seven patients received a single dose of bevacizumab, including 13 patients from China and 14 patients from Italy, among whom 1 patient dropped out owing to the withdrawal of consent (Fig. 1). The median age of patients was 62 years, and 20 (77%) were men. The median time from symptom onset to admission was 10 days, while the median interval from admission to bevacizumab administration was 7 days. Fever 25 (96%), dry cough 21 (81%), shortness of breath 19 (73%), and fatigue 17 (65%) were common onset symptoms. Thirteen patients had hypertension history (50%) and six (23%) had diabetes history. Cofounding factors including age, maximum body temperature, smoking, asthma, hypertension, diabetes, cardiovascular

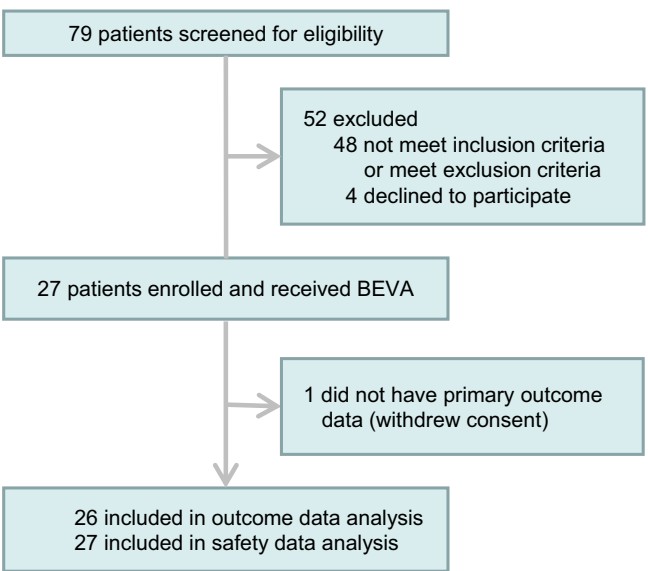

**Fig. 1 CONSORT flow diagram.** Twenty-seven participants were enrolled. All participants received bevacizumab treatment (BEVA). One participant dropped out.

disease, chronic obstructive pulmonary disease (COPD), and common clinical symptoms were indistinguishable between Chinese and Italian patients. Days from hospitalization to bevacizumab treatment were significantly longer for patients in the Chinese group than the Italian group (Table 1). Notably, a significant difference in BMI existed between Italian patients and Chinese patients.

**$PaO_2/FiO_2$ improvement**. $PaO_2/FiO_2$ values markedly increased at days 1 and 7 after bevacizumab administration compared to the baseline values. Differences in genetic background, treatment approaches, and admission-to-bevacizumab administration time between the Chinese and Italian populations should not be ignored. Therefore, we performed subgroup analysis. Both populations showed a marked increase of $PaO_2/FiO_2$ values at day 1 post-bevacizumab treatment relative to the baseline values, whereas, at day 7, only the Italian population displayed increases of $PaO_2/FiO_2$ ratios with statistical significance (Fig. 2).

**Oxygen-support status**. After receiving a single dose of bevacizumab, 24 of 26 patients (92%) showed improvement and 2 patients (8%) showed no change in oxygen-support within 28-day follow-up. Inspiringly, 17 of 26 patients (65%) were discharged. Six patients receiving invasive or non-invasive ventilation stopped ventilator support and 4 of 6 patients (67%) were discharged within follow-up. For the 2 patients with no change in oxygen-support, one breathed ambient air at baseline. None of the 26 patients showed worsened status of oxygen-support nor died after bevacizumab treatment in the 28-day follow-up trial (Fig. 3).

**Chest radiological imaging**. Eight patients received chest CT scanning and 12 patients received bedside or traditional X-ray examination alternatively at the required time points. Six patients, despite good recovery (patient nos. 15, 19, 21, 22, 23, and 26 shown in Fig. 3), were unable to receive chest radiological examination in time due to unavailability of CT machines, shortage of medical staff, or being discharged. The quantified results of chest CT showed that the total lesion areas ($cm^3$) and the lesion ratios (%) for both the lungs significantly reduced at day 7 relative to baseline. Regarding changes in characteristic

**Table 1 Baseline demographic and clinical characteristics of the patients.**

| Characteristics | BEVA ($n = 26$) | Control ($n = 26$) | p- value | BEVA | | |
| --- | --- | --- | --- | --- | --- | --- |
| | | | | Chinese ($n = 12$) | Italian ($n = 14$) | p- value |
| Age, median [IQR], years | 62 (55, 66) | 65 (58, 72) | 0.131 | 62 (53, 67) | 63 (59, 66) | 0.502 |
| Maximum body temperature, median [IQR], °C | 38.5 (38.0, 39.0) | 38.4 (37.8, 38.8)[a] | 0.223 | 38.7 (38.3, 39.5) | 38.4 (38.0, 39.0) | 0.407 |
| Symptom onset to admission, median [IQR], days | 10 (7, 15) | 10 (6, 15)[b] | 0.887 | 13 (9, 16) | 9 (6, 11) | 0.044 |
| Admission to BEVA treatment, median [IQR], days | 7 (3, 12) | 4 (0, 11) | 0.074 | 12 (9, 24) | 4 (3, 6) | 0.0003 |
| Sex, no. (%) | | | 0.532 | | | 0.065 |
| Female | 6 (23) | 8 (31) | | 5 (42) | 1 (7) | |
| Male | 20 (77) | 18 (69) | | 7 (58) | 13 (93) | |
| BMI, median [IQR], kg/m$^2$ | 24.6 (23.2, 28.1) | 25.0 (22.1, 27.0) | 0.673 | 23.2 (21.7, 24.0) | 26.5 (25.0, 31.0) | 0.001 |
| Smoke history, no. (%) | 7 (27) | 7 (27) | >0.999 | 1 (8) | 6 (43) | 0.081 |
| Medical history, no. (%) | | | | | | |
| Heart disease | 2 (8) | 3 (12) | >0.999 | 0 (0) | 2 (14) | 0.483 |
| Hypertension | 13 (50) | 14 (54) | 0.781 | 4 (33) | 9 (64) | 0.238 |
| Diabetes | 6 (23) | 5 (19) | 0.734 | 2 (17) | 4 (29) | 0.652 |
| COPD | 3 (12) | 4 (15) | >0.999 | 1 (8) | 2 (14) | >0.999 |
| Asthma | 2 (8) | 2 (8) | >0.999 | 0 (0) | 2 (14) | 0.483 |
| Symptoms, no. (%) | | | | | | |
| Fever | 25 (96) | 23 (89) | 0.610 | 12 (100) | 13 (93) | >0.999 |
| Dry cough | 21 (81) | 15 (58) | 0.071 | 8 (67) | 13 (93) | 0.148 |
| Expectoration | 7 (27) | 7 (27) | >0.999 | 4 (33) | 3 (21) | 0.665 |
| Fatigue | 17 (65) | 10 (39) | 0.052 | 8 (67) | 9 (64) | >0.999 |
| Shortness of breath | 19 (73) | 11 (42) | 0.025 | 6 (50) | 13 (93) | 0.026 |
| Chest distress | 5 (19) | 7 (27) | 0.510 | 4 (33) | 1 (7) | 0.148 |
| Chill | 6 (23) | 1 (4) | 0.099 | 3 (25) | 3 (21) | >0.999 |
| Headache | 6 (23) | 2 (8) | 0.248 | 2 (17) | 4 (29) | 0.652 |

*IQR* interquartile range, *BEVA* bevacizumab, *BMI* body mass index, *COPD* chronic obstructive pulmonary disease; *n* = sample size; Wilcoxon signed-rank test, and Chi-square test were used to calculate p-values. *p* < 0.05 for two-tailed hypothesis tests was considered statistically significant.
[a]Four data were not recorded.
[b]One datum was not recorded. Source data are provided as a Source Data file.

pulmonary lesions, the number of ground-glass opacities increased and the number of patchy shadows decreased at day 7 compared to the baseline value, which indicated the pulmonary inflammatory exudation was progressively absorbed, however, the increase and decrease were not statistically significant. No consolidation lesions were shown in chest CT images of these patients. The semi-quantification of chest X-ray images showed, the lesion ratios of the right lungs remarkably reduced at days 3 and 7 compared to baseline (Table 2). The representative chest CT as well as X-ray images were shown in Figs. 4 and 5.

**Fever symptom**. Among the 26-treated patients, 14 had a fever at the time prior to bevacizumab treatment and 12 had no fever. Surprisingly, we observed a phenomenon of rapid abatement of fever in 13 of 14 febrile patients (93%) within 3 days after bevacizumab treatment regardless of the days of febrile from admission to bevacizumab treatment, which were in a range of 0–8 days (Fig. 6 and Supplementary Fig. 1). One trachea intubation patient was an exception with a persistent fever, who had got sepsis that was evidenced by blood and urine bacteria culture. From the available values of laboratory tests at baseline and day 7, the peripheral blood lymphocyte counts and the level of CRP were significantly increased and decreased at day 7, respectively (Fig. 6 and Supplementary Table 1).

**Adverse events**. Safety data were collected from all 27 patients and were assessed by the Common Terminology Criteria for Adverse Events (CTCAE). Elevation of hepatic (30%) including alanine aminotransferase (ALT) and aspartate aminotransferase (AST) was the most common adverse event. According to CTCAE, ALT grade 1: 4 patients (15%); ALT grade 2: 2 patients (7%); ALT grade 3: 2 patients (7%); AST grade 1: 7 patients (25%) and AST grade 3: 1 patient (4%). Other adverse events included: 5 patients (19%) showed reduced hemoglobin values (grade 1); 4 patients (15%) with decreased platelet counts (grade 1); 3 patients (11%) showed modest elevation of blood pressure (grade 2), a common adverse event related to bevacizumab; 2 patients (7%)

had elevated blood urea nitrogen; 2 patients (7%) developed sepsis (grade 1 and 3). The following adverse events were observed in 1 patient (4%) each: hemorrhagic urea (grade 1), diarrhea (grade 1), skin rash (grade 2), muscle pains in the lower extremity (grade 1), superficial phlebitis at the site of venepuncture (grade 2), and sinus tachycardia with atrial premature beats occurred in one patient (grade 1) (Table 3). Of note, some of these adverse events might not directly be related to bevacizumab treatment because other factors such as antiviral drugs, Chinese herb medicine, genetic variations, and concomitant disease could potentially contribute to the development of these adverse events.

**External controls**. For the external comparison cohort, we conducted retrospective screening of patients with severe Covid-19 who had complete data set of the $PaO_2/FiO_2$ in the same center within a similar timeframe (±5 days), i.e., between the 10th of February and the 13th of March of 2020 in China and between the 20th of March and the 10th of the April of 2020 in Italy. The complete data set of the $PaO_2/FiO_2$ referred to complete $PaO_2/FiO_2$ data records of days 0, 1, and 7. If the patients died during this period, the records of $PaO_2/FiO_2$ data should be complete before the patient's death. The inclusion criteria between the external control and bevacizumab-treated groups were similar (see "Methods" section for details). A total number of 137 patients with severe Covid-19 were deemed to be eligible for external controls. Among them, 101 patients were excluded for the following reasons: (1) for not meeting the inclusion criteria; and (2) other disorders that might affect respiratory outcomes and survival. Ultimately, 36 patients were selected as potential controls. Owing to the lack of complete data sets, 10 patients were further excluded. As a result, 26 patients were used as an external control group (Supplementary Fig. 2). The Ethical committees in both centers approved the observational data collection of patients with Covid-19 for the purpose as controls. The baseline characteristics were comparable between the two groups (Table 1). Regarding the prognosis of external controls, 3 patients

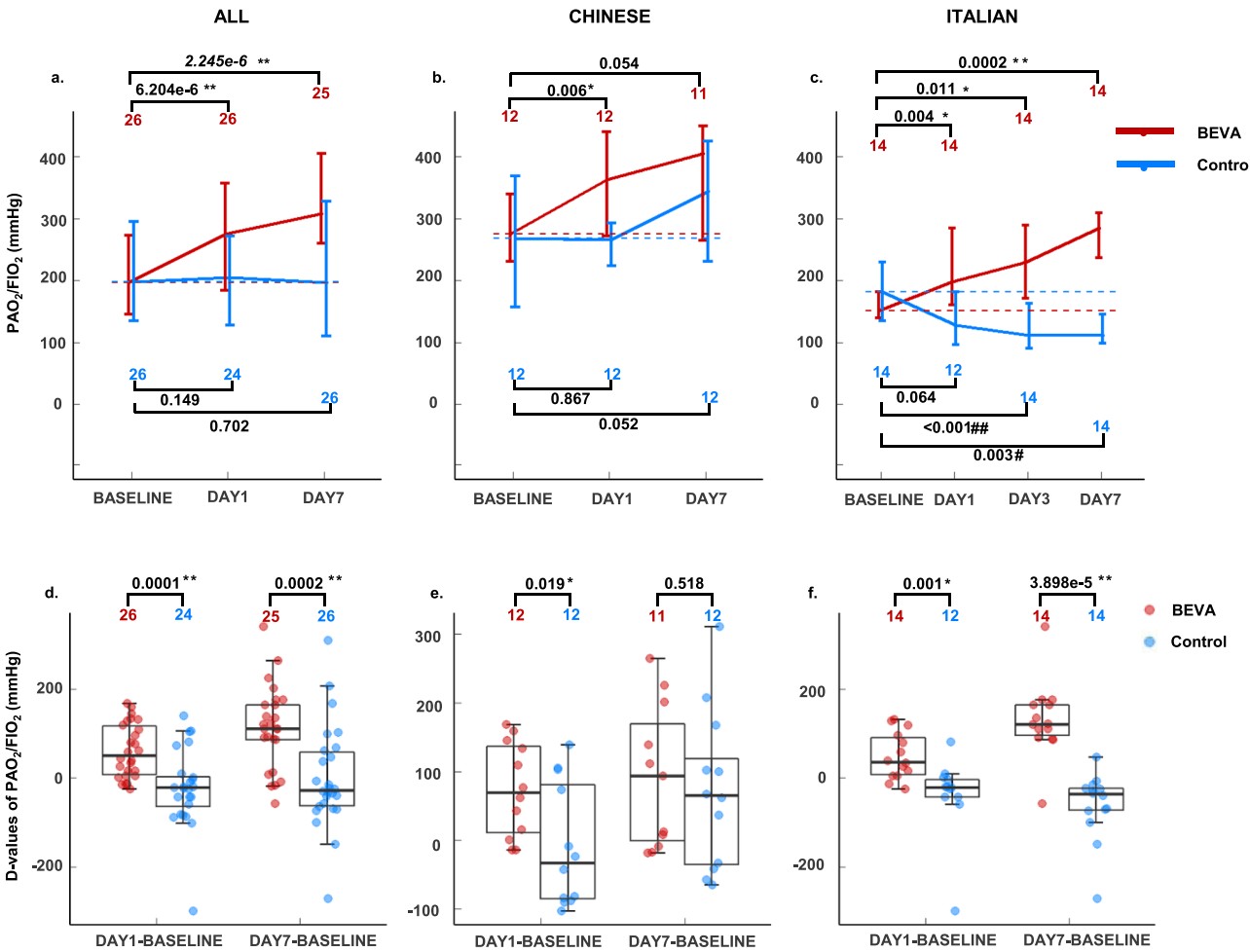

**Fig. 2 Comparison of dynamic changes of PaO$_2$/FiO$_2$ between BEVA (bevacizumab) and control groups.** PaO$_2$/FiO$_2$ values in all patients (**a**), Chinese (**b**), and Italian (**c**) groups. The dashed lines represent the baselines; nodes in the lines represent the median values; the lower and upper values of the bar correspond to the 25th and 75th percentiles (Q1 and Q3); the length of the bar represents the interquartile range (IQR); red and blue numbers correspond to the sample sizes (*n*); numbers on the horizontal lines represent *p*-values. *D*-values (the difference values between days 1 or 7 and baseline) of individual patients were utilized for the comparisons between BEVA and control groups of all patients (**d**), Chinese (**e**), and Italian (**f**) patients. The center lines of boxes represent median values; the lower and upper hinges represent Q1 and Q3, the range between which represents IQR; whiskers correspond to the highest or lowest values of non-outlier data (within 1.5 × IQR from the lower or upper hinges); numbers on the horizontal lines represent *p*-values; red and blue numbers correspond to the sample sizes (*n*); the red and blue dots represent the data of individual patients. For **a**–**c**, *p*-values were calculated by Wilcoxon matched-pairs signed-rank test. For **d**–**f**, *p*-values were calculated by Wilcoxon signed-rank test. PaO$_2$/FiO$_2$ partial arterial oxygen pressure to fraction of inspiration O$_2$ ratio. *p* < 0.05 for two-tailed hypothesis tests was considered statistically significant. Source data are provided as a Source Data file.

died of Covid-19 and 0 patient died in the bevacizumab-treated group (11.5% vs. 0%) (Supplementary Fig. 3).

Compared with the control group, bevacizumab treatment significantly elevated PaO$_2$/FiO$_2$ values on days 1 and 7 (Fig. 2 and Supplementary Fig. 4). Of note, day 3 data from the Wuhan site was not collected due to a lack of sufficient medical facilities and medical personals. It seemed that standard care had no impact on improving PaO$_2$/FiO$_2$ values on average in control groups during the comparable period. The basic regimens for treating patients with severe Covid-19 were similar in the Wuhan and Lombardian sites, which included antiviral drugs, hydroxy-chloroquine, antibiotics, steroids, antipyretics, and supportive care. However, there were differences between the two sites. At the Lombardian site, all patients in the treated and control groups received treatment with anticoagulants, whereas only 2 patients at the Wuhan site received anticoagulants. According to the standard care recommended by the Health Ministry of China, Chinese herb medicine was used for treating all patients with

severe Covid-19, and Chinese traditional medicine was not used at the Lombardian site.

Relative to controls, the beneficial effect of bevacizumab on oxygenation became obviously improved after 24-h bevacizumab treatment. These findings indicate that the clinical benefits of bevacizumab treatment occur already at early time points. Compared with the control group, a substantially increased number of patients improved the 28-day oxygen-support status (control, 62% vs. bevacizumab, 92%) and the discharge rate (control, 46% vs. bevacizumab, 65%) (Supplementary Fig. 3). The bevacizumab-treated patients also showed less deterioration of oxygen-support status relative to the control group (control, 19% vs. bevacizumab, 0%). Additionally, bevacizumab treatment significantly shortened the duration of oxygen-support (control, median 20 [interquartile range (IQR) 14,28] days vs. bevacizu-mab, median 9 [IQR 5,19] days, *p* = 0.003). When compared our data with other published studies (Supplementary Table 2), bevacizumab showed potential comparably competitive strength

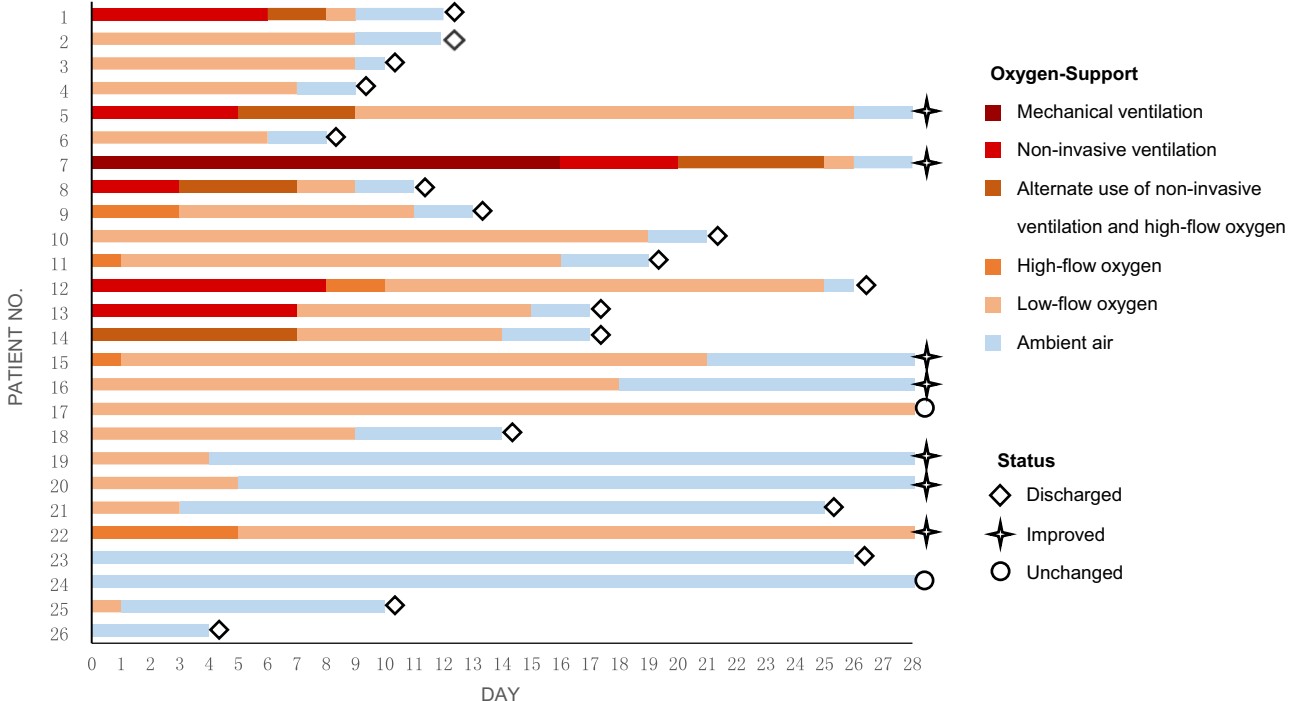

**Fig. 3 Changes in oxygen-support status in individual patients.** Baseline (day 0) is the day when treatment with a single dose of bevacizumab was performed. The final oxygen-support statuses are on the discharge date or at the end of the 28-day follow-up. For each individual patient, colored columns represent the oxygen-support status of the patient over time. Diamond symbols represent patients discharged from hospitals; star symbols represent patients improved but not discharged, and circle symbols show patients unchanged.

to remdesivir, lopinavir–ritonavir, and convalescent plasma. Thus, our findings are encouraging and may become a first-line therapeutic regimen for treating patients with severe Covid-19.

We established thresholds of $PaO_2/FiO_2$ further assessing the beneficial effect of bevacizumab treatment compared with external controls. We defined thresholds of $PaO_2/FiO_2$ increases by 50 and by 100 mmHg from the baseline for days 1 and 7, respectively. On day 7, according to the stratification of ARDS as indicated above, achieving these thresholds means to gain one class from moderate ($PaO_2/FiO_2$ 100–200 mmHg) to mild ($PaO_2/FiO_2$ 200–300 mmHg), which was an excellent proxy of clinical global prognosis. An increase of $PaO_2/FiO_2$ by 100 mmHg likely presented healing and shortness of hospitalization, and most of all, reducing the possibility of intubation and receiving further treatment in ICU. Optimization of resources, limitation of ICU beds, and sustainability were crucial clues for our hospitals, especially in a marked situation such as Covid-19 pandemics. After establishing thresholds, the results showed that 13 patients (50%) on day 1 and 15 patients (57.7%) on day 7 in the bevacizumab-treated group reached the thresholds whereas only 5 patients (19.2%) on day 1 and 4 patients (15.4%) on day 7 reached the threshold in the control group (Supplementary Fig. 5). The net increases on day 1 were by 30.8% and day 7 by 42.3%, which were considered to substantial increases.

## Discussion

Despite the initiation of numerous trials since the Covid-19 outbreak, almost no effective therapy is available. In this pre-liminary trial, we report clinical findings in patients with severe Covid-19 from two medical centers of China and Italy. To the best of our knowledge, the concept of implementing bevacizumab for treating patients with severe Covid-19 is completely novel and has not been explored.

A single-dose treatment with bevacizumab improved the oxygen-support status in 92% of patients during 28-day follow-up without causing death, whereas in the external comparison cohort receiving only standard care, the improvement rate of oxygen-support status was only 62%, and the deterioration of oxygen-support status rate was 19% including three death. Bevacizumab treatment markedly shortened the duration of oxygen-support compared with the control cohort.

There are several unexpected findings from this trial: (1) Rapid improvement of the $PaO_2/FiO_2$ values. Within 24 h after delivery of bevacizumab, the majority of patients with severe Covid-19 showed rapid improvement of the $PaO_2/FiO_2$ ratio; (2) Rapid abatement of fever. Within 72 h after bevacizumab, 93% of patients exhibited normal body temperature; Although limited laboratory tests were available, the following two findings are still noteworthy: (3) Increase of peripheral blood (PB) lymphocyte counts. Patients suffered from lymphopenia, and on day 7 after bevacizumab treatment, they showed marked increases of PB lymphocytes; and (4) Anti-inflammation. An over 12-fold decrease of CRP was observed in bevacizumab-treated patients.

Rapid improvement of $PaO_2/FiO_2$ values might reflect the anti-vascular leakiness effect by bevacizumab. A similar rapid response for improving visual acuity by anti-VEGF treatment was also seen in patients with wet age-related macular degeneration (AMD)[25]. VEGF likely contributed to Covid-19-induced inflammation. It is known that VEGF mobilizes inflammatory cells to pathological tissues and anti-VEGF would alleviate fever by antagonizing the virus-triggered inflammation[26,27]. At this time of writing, the mechanism underlying increases of lymphocytes is unknown. It is speculated that blocking VEGF by bevacizumab might affect extravasation and redistribution of lymphocytes. As lymphopenia is one of the key pathological changes in contributing to Covid-19 fatality[28,29], increases in the lymphocytic population would reduce the death rates in severe patients.

**Table 2 Chest radiological imaging quantification.**

|  | n | Baseline | Day 3 | p-value | Day 7 | p-value |
|---|---|---|---|---|---|---|
| Chest CT quantification |  |  |  |  |  |  |
| Patchy shadow, median [IQR], numbers | 8 | 2.5 (0, 4.0) | – | – | 1.5 (0, 3.5) | 0.641 |
| Ground-glass opacity, median [IQR], numbers | 8 | 3.5 (1.5, 7.0) | – | – | 6.5 (3.5, 10.5) | 0.094 |
| Total Lesion area, median [IQR], cm$^3$ | 8 | 1055.8 (532.8, 1409.4) | – | – | 680.7 (395.9, 926.2) | 0.039 |
| Lesion ratio, median [IQR], %—right lung | 8 | 19.2 (12.8, 33.6) | – | – | 12.4 (9.7, 15.6) | 0.023 |
| Lesion ratio, median [IQR], %—left lung | 8 | 35.7 (25.0, 45.2) | – | – | 17.7 (13.0, 31.8) | 0.016 |
| Chest X-ray semi-quantification |  |  |  |  |  |  |
| Lesion ratio, median [IQR], %— right lung | 12 | 48.5 (30.7, 71.8) | 34.0 (23.4, 50.7) | 0.024 | 26.7 (4.6, 36.4) | 0.005 |
| Lesion ratio, median [IQR], %— left lung | 12 | 47.6 (39.4, 72.8) | 39.8 (30.8, 49.6) | 0.077 | 28.1 (18.8, 55.2) | 0.052 |

Chest radiological imaging quantification results are based on available images. Chest CT images on baseline and day 7 post-bevacizumab treatment were from 6 Chinese patients and 2 Italian patients. Chest X-ray images on baseline, days 3 and 7 post-bevacizumab treatment were from 12 Italian patients. Patient nos. 15, 19, 21, 22, and 23 were unable to receive chest radiological examination on required time points, although they recovered well (shown in Fig. 3). Patient no. 26 was discharged on day 4. *CT* computed tomography, *IQR* interquartile range; Wilcoxon matched-pairs signed-rank test was used to calculate the p-values between day 3 and baseline, and day 7 and baseline. $p < 0.05$ for two-tailed hypothesis tests was considered statistically significant. Source data are provided as a Source Data file.

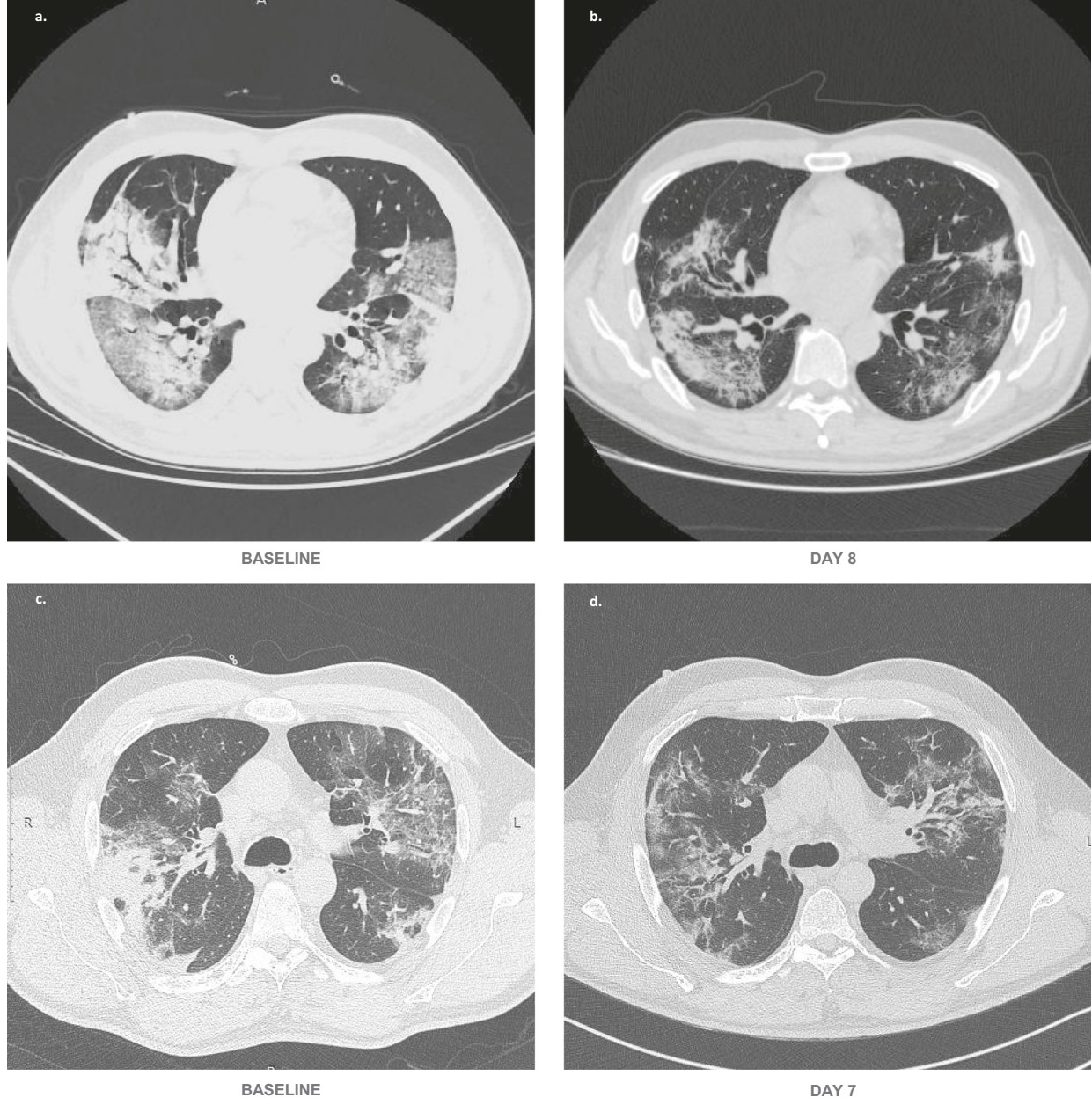

**Fig. 4 Representative chest CT images.** Typical chest CT images of a Chinese patient with severe Covid-19 who received bevacizumab at day 8 (**b**) relative to the prior treatment baseline (**a**). Typical chest CT images of an Italian patient with severe Covid-19 who received bevacizumab at day 7 (**d**) relative to the prior treatment baseline (**c**).

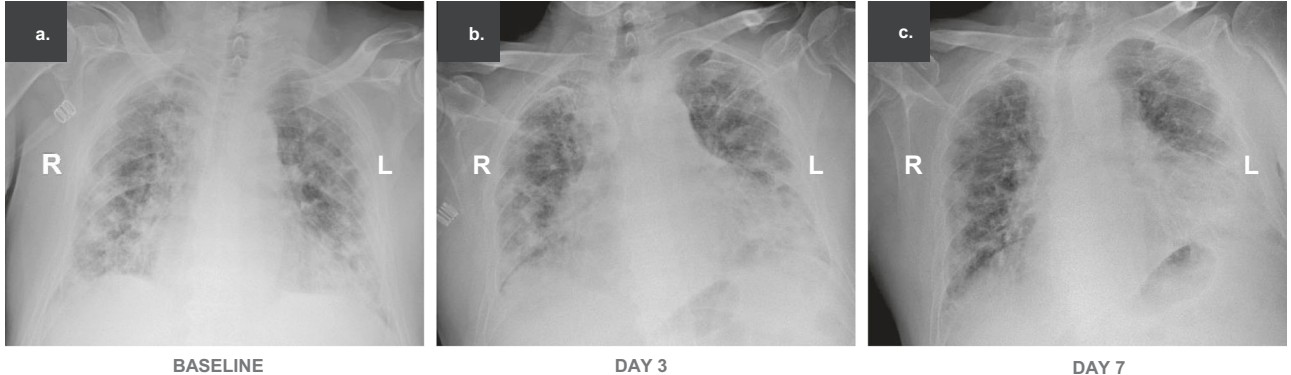

**Fig. 5 Representative chest X-ray images.** Chest X-ray images of an Italian patient with severe Covid-19 who received bevacizumab treatment at days 0 (**a**), 3 (**b**), and 7 (**c**) relative to the prior treatment baseline.

**Fig. 6 Changes in fever symptom, lymphocyte counts, and inflammation markers of individual patients.** Dynamic changes in fever status of 14 BEVA (bevacizumab)-treated patients who had a fever at enrollment (**a**). Day 0 marks the time point for initiating bevacizumab treatment; red, orange, and blue columns indicate the duration of fever or normal body temperature status before and after BEVA treatment; and diamonds represent discharge. Numbers along the vertical axis represent the patient numbers that match Fig. 2. Changes of lymphocyte counts (*n* = 14) (**b**) and C-reactive protein (CRP) levels (*n* = 9) (**c**) before and after BEVA treatment were analyzed by the Wilcoxon signed-rank test. The center lines of boxes represent median values; the lower and upper hinges represent the 25th and 75th percentiles (Q1 and Q3), the range between which represents the interquartile range (IQR); whiskers correspond to the highest or lowest values of non-outlier data (within 1.5 × IQR from the lower or upper hinges); numbers on the horizontal lines represent *p*-values. The red dots represent the data of individual treated patients. *p* < 0.05 for two-tailed hypothesis tests was considered statistically significant. Source data are provided as a Source Data file.

**Table 3 Summary of adverse events.**

| Events Adverse events (%) | Chinese (n = 13) | | | Italian (n = 14) | | | All (n = 27) | | | Total |
|---|---|---|---|---|---|---|---|---|---|---|
| | Grade 1 | Grade 2 | Grade 3 | Grade 1 | Grade 2 | Grade 3 | Grade 1 | Grade 2 | Grade 3 | |
| Non-bevacizumab-related adverse effects (%) | | | | | | | | | | |
| ALT | 3 | | 1 | 1 | 2 | 1 | 4 | 2 | 2 | 8 |
| AST | 4 | | | 3 | | 1 | 7 | | 1 | 8 |
| Low hemoglobin | 3 | | | 2 | | | 5 | | | 5 |
| Low platelets | 4 | | | | | | 4 | | | 4 |
| High blood urea nitrogen | NA | NA | NA | NA | NA | NA | NA | NA | NA | 2 |
| Skin rash | 1 | | | | | | 1 | | | 1 |
| Local superficial phlebitis | | | | | 1 | | | 1 | | 1 |
| Sepsis | 1 | | | | | 1 | 1 | | 1 | 2 |
| Bevacizumab-related adverse effects (%) | | | | | | | | | | |
| Hypertension (grade ≥ 2) | | 2 | | | 1 | | | 3 | | 3 |
| Muscle pains in lower extremity | 1 | | | | | | 1 | | | 1 |
| Diarrhea | 1 | | | | | | 1 | | | 1 |
| Cardiovascular events | 1 | | | | | | 1 | | | 1 |
| Hematuria | 1 | | | | | | 1 | | | 1 |

Adverse events of bevacizumab-treated patients were presented according to the Common Terminology Criteria for Adverse Events (CTCAE) Version 5.0.
ALT alanine aminotransferase, AST aspartate aminotransferase, NA not applicable.

Bevacizumab at a dose range of 5–15 mg/kg is routinely used in oncology and about 7.5 mg/kg used in our trial was within the lower range. Bevacizumab-related serious adverse events such as gastrointestinal perforation, hemorrhages, and the nephrotic syndrome were absent in both bevacizumab-treated Chinese and Italian populations. No severe safety concerns were detected in this drug use cohort.

It is known that circulating VEGF levels were elevated in patients with severe Covid-19 in Wuhan[5]. Owing to the following reasons, we did not measure plasma VEGF levels: (1) the Covid-19 pandemic-associated limitation of feasibility; (2) plasma VEGF levels may not reflect local VEGF production in pulmonary tissues; and (3) most VEGF isoforms bind to heparin and may be sequestered the tissues where they produce.

The limitations of these results include the non-randomized and uncontrolled nature of this trial, the short term follow-up, the high and unexplained variation in the baseline of $PaO_2/FiO_2$ between the two centers, and the small size of the cohort. However, improvements of multiple clinical parameters in bevacizumab-treated patients suggest that anti-VEGF might benefit for treating patients with Covid-19. In particular, it is worthy to design future trials by combining bevacizumab with other therapeutic modalities such as antiviral and anti-inflammatory drugs. Given the limited medical supply and available facility in most medical centers, a single injection of bevacizumab might immediately relieve clinical symptoms and early discharge of patients with severe Covid-19. The therapeutic benefits of bevacizumab monotherapy and combination therapy warrant future validation in patients with severe Covid-19 by randomized and placebo-controlled trials.

## Methods

**Study participants.** Patients aged 18–80 years with a confirmed Covid-19 diagnosis were eligible if they had respiratory distress with a respiratory rate (RR) of ≥30 times/min, oxygen saturation (SpO$_2$) of ≤93% while they were breathing ambient air, or a partial arterial oxygen pressure to the fraction of inspiration O$_2$ ratio (PaO$_2$/FiO$_2$) of >100 and ≤300 mmHg, and diffuse pneumonia confirmed by chest radiological imaging. A confirmed Covid-19 diagnosis was based on epidemiological history (including cluster transmission) and a positive reverse-transcriptase polymerase chain reaction (RT-PCR) assay (BioPerfectus Technologies, China; ELITech Group, France; Seegene, Korea) performed by the local center for disease control or a designated diagnostic laboratory.

The following patients were excluded from the trial: (1) patients with severe hepatic dysfunction (Child-Pugh score ≥ C or aspartate aminotransferase level >5 times the upper reference limit, URL); (2) patients with severe renal dysfunction (estimated glomerular filtration rate ≤30 mL/min/1.73 m$^2$) or who required continuous renal replacement therapy, haemodialysis, or peritoneal dialysis; (3) patients with uncontrolled hypertension (sitting systolic blood pressure > 160 mmHg or diastolic blood pressure >100 mmHg) or a history of hypertension crisis or hypertensive encephalopathy; (4) patients with poorly controlled heart diseases, such as New York Heart Association class II or higher cardiac insufficiency, unstable angina pectoris, myocardial infarction within 1 year before enrollment, or supraventricular or ventricular arrhythmia needing treatment or intervention; (5) patients with hereditary bleeding tendency or coagulopathy, and patients who received full-dose anticoagulant or thrombolytic therapy within 10 days before enrollment, or non-steroidal anti-inflammatory drugs with platelet suppression within 10 days before enrolment (except those who used small doses of aspirin [≤325 mg/day] for preventive use); (6) patients with thrombosis within 6 months before enrollment, patients who had experienced arterial/venous thromboembolic events, such as ischemic stroke, transient ischemic attack, deep venous thrombosis, or pulmonary embolism within 1 year prior to trial enrollment, and patients with severe vascular diseases (including aneurysms or arterial thrombosis requiring surgery) within 6 months before trial enrollment; (7) patients with unhealed wounds, active gastric ulcers, or fractures; patients with gastrointestinal perforation, gastrointestinal fistula, abdominal abscess, or visceral fistula formation within 6 months before trial enrollment; (8) patients who had undergone major surgery (including preoperative chest biopsy) or received major trauma (such as a fracture) within 28 days before enrollment; patients who might need surgery during the trial; (9) patients with severe, active bleeding such as haemoptysis, gastrointestinal bleeding, central nervous system bleeding, and epistaxis within 1 month before trial enrollment; (10) patients with malignant tumours within 5 years before trial enrollment; (11) patients allergic to bevacizumab or its components; (12) patients with untreated active hepatitis or human immunodeficiency virus (HIV)-positive patients; pregnant and lactating women and those planning to get pregnant; (13) patients who participated in other clinical trials or not considered suitable for this trial by the researchers; or (14) patients who did not provide signed informed consent. The study protocol is provided in Supplementary Note 1.

**Control cohort.** An external comparison cohort was selected from the same center and the patients met the inclusion criteria without severe hepatic, renal or cardiac dysfunction, and were comparable to the bevacizumab-treated groups. For the selection area of controls, we screened from (1) other wards within the same center; and (2) the patients who declined to sign the informed consent to receive bevacizumab treatment but eligible for controls in the designated wards within the same center. Days 1 and 7 in control groups were defined to be comparable to the treated group timeframe for each cohort. The period from the admission time to day 0 was comparable between the external control group and the bevacizumab-treated group in the same center. The detailed information on the identification and selection of external controls were described in the Results section.

Inclusion criteria for the external control group were as follows: Patients aged 18–80 years with a confirmed Covid-19 diagnosis were eligible if they had respiratory distress with a respiratory rate (RR) of ≥30 times/min, oxygen saturation (SpO$_2$) of ≤93% while breathing ambient air, the PaO$_2$/FiO$_2$ ratio of >100 and ≤300 mmHg, and diffuse pneumonia confirmed by chest radiological imaging. A confirmed Covid-19 diagnosis was based on epidemiological history (including cluster transmission) and a positive reverse-transcriptase polymerase chain reaction (RT-PCR) assay (BioPerfectus Technologies, China; ELITech

Group, France; Seegene, Korea) performed by the local center for disease control or a designated diagnostic laboratory. Of note, 3 patients were included as controls: 1 with favism; 1 received anticoagulant; and 1 had prostate cancer 2 years ago. During the trial period, their disorders remained stable and well-manageable. As determined by experienced doctors and clinical research experts, these three patients were eligible to serve as controls.

The detailed information of exclusion criteria for the external control group is also described in the "Methods" section of the revised manuscript. Exclusion criteria for the external control group: (1) patients with severe hepatic dysfunction (Child-Pugh score ≥ C or aspartate aminotransferase level >5 times the upper reference limit, URL); (2) patients with severe renal dysfunction (estimated glomerular filtration rate ≤30 mL/min/1.73 m²) or who required continuous renal replacement therapy, haemodialysis, or peritoneal dialysis; (3) patients with uncontrolled hypertension (sitting systolic blood pressure >160 mmHg or diastolic blood pressure >100 mmHg) or a history of hypertension crisis or hypertensive encephalopathy; (4) patients with poorly controlled heart diseases, such as New York Heart Association class II or higher cardiac insufficiency, unstable angina pectoris, myocardial infarction within 1 year before enrollment, or supraventricular or ventricular arrhythmia needing treatment or intervention; (5) patients with hereditary bleeding tendency or coagulopathy, and patients who received full-dose anticoagulant or thrombolytic therapy within 10 days before enrollment, or non-steroidal anti-inflammatory drugs with platelet suppression within 10 days before enrolment (except those who used small doses of aspirin [≤325 mg/day] for preventive use); (6) patients with thrombosis within 6 months before enrollment, patients who had experienced arterial/venous thromboembolic events, such as ischemic stroke, transient ischemic attack, deep venous thrombosis, or pulmonary embolism within 1 year prior to trial enrollment, and patients with severe vascular diseases (including aneurysms or arterial thrombosis requiring surgery) within 6 months before trial enrollment; (7) patients with unhealed wounds, active gastric ulcers, or fractures; patients with gastrointestinal perforation, gastrointestinal fistula, abdominal abscess, or visceral fistula formation within 6 months before trial enrollment; (8) patients who had undergone major surgery (including preoperative chest biopsy) or received major trauma (such as a fracture) within 28 days before enrollment; patients who might need surgery during the trial; (9) patients with severe, active bleeding such as haemoptysis, gastrointestinal bleeding, central nervous system bleeding, and epistaxis within 1 month before trial enrollment; (10) patients with malignant tumours within 5 years before trial enrollment; (11) patients with untreated active hepatitis or human immunodeficiency virus (HIV)-positive patients, pregnant and lactating women; (12) patients who participated in other clinical trials.

**Trial design and oversight**. This was a single-arm trial (ClinicalTrials.gov NCT04275414) conducted from February 15 to April 5, 2020 (the date of enrolment of the last patient), which enrolled patients in the Renmin Hospital of Wuhan University, Wuhan, Hubei Province, China from February 15 to March 8, 2020, and enrolled patients in Italia Hospital S.p.A. Ospedale Generale di Zona Moriggia – Pelascini, Gravedona ed Uniti (CO), Italy from March 25 to April 5, 2020, and followed up for 28 days or until hospital discharge. All eligible patients evaluated by investigators were enrolled. Each eligible patient received a single dose (500 mg) of bevacizumab (Qilu Pharmaceutical Co. LTD and Roche Pharmaceutical Co. LTD) dissolved in 100 mL of saline intravenously in no <90 min under electro-cardiography monitoring and standard care. The trial was approved by the ethics committees of Qilu Hospital of Shandong University, Renmin Hospital of Wuhan University, and Italia Hospital S.p.A. Ospedale Generale di Zona Moriggia – Pelascini, Gravedona ed Uniti (CO). Written informed consents were obtained from all patients or the legal representatives of patients if they were unable to provide consents by themselves. The adverse events were monitored and adjudicated by the Safety Monitoring Committee. All the adverse events were handled timely with proper medical treatment to avoid further damage. The trial was conducted in accordance with the principles of the Declaration of Helsinki and the Good Clinical Practice guidelines of the International Conference on Harmonisation.

**Standard care**. Standard care included supplemental oxygen, non-invasive and invasive ventilation, antivirotic or antibiotic agents, vasopressor support, and extracorporeal membrane oxygenation as necessary. The basic regimens for treating patients with severe Covid-19 were similar in the Wuhan and Lombardian sites, which included antiviral drugs, hydroxychloroquine, antibiotics, steroids, antipyretics, and supportive care. However, there were differences between the two sites. For steroids, 15 of 26 bevacizumab-treated patients received steroid treatment, 7 Italian patients, and 8 Chinese patients. In Chinese control patients, 8 patients received steroids. In Italian control patients, 5 patients received steroids. For anticoagulants, at the Lombardian site, all patients received treatment with enoxaparin, whereas only 2 patients at the Wuhan site received enoxaparin. For control patients, 4 Chinese and 6 Italian patients received enozaparin treatments. According to the standard care recommended by the Health Ministry of China, Chinese herb medicine was used for treating all Wuhan patients with severe Covid-19, and Chinese traditional medicine was not used at the Lombardian site.

**Outcomes and measures**. Primary outcomes were changes of $PaO_2/FiO_2$ at days 1 and 7. $PaO_2$ was measured by arterial blood gas (ABG) and $FiO_2$ at the time of

clinical ABG was obtained. Secondary outcomes included the change of chest radiological imaging on day 7, as well as oxygen-support status, discharge rate, and change of fever symptom during 28 days follow-up. Oxygen-support status referred to mechanical ventilation, non-invasive ventilation, a transition status of alternate use of non-invasive ventilation and high-flow oxygen, high-flow oxygen, low-flow oxygen, or ambient air. Chest radiological imaging referred to chest CT or X-ray, which was quantified or semi-quantified by software or a team of experts, respectively. Fever was defined as axillary temperature of ≥37.5 °C. All measurements were taken from distinct patients.

**Disease severity classification**. The severity of Covid-19 was defined according to the Chinese Clinical Guidance for Covid-19 Diagnosis and Treatment (the 7th edition). According to this guidance, Covid-19 patients were triaged into the following categories: (1) severe type was defined as adults meeting any of the following criteria: a. shortness of breath, RR ≥ 30 times/min; b. oxygen saturation ($SpO_2$) ≤93% at rest; c. $PaO_2/FiO_2$ ≤ 300 mmHg, patients whose pulmonary imaging displays accelerated progression of lesion >50% within 24–48 h should be treated as severe type; (2) critically severe type, patients met any of the following: a. respiratory failure requiring mechanical ventilation; b. shock; c. critical comorbidity with other organ failures that need intensive care unit (ICU) monitoring and treatment.

**Clinical, laboratory, and radiological data collection**. Arterial blood gas analysis, chest CT scanning, chest X-ray, and laboratory tests were performed in Renmin Hospital of Wuhan University, and Moriggia – Pelascini Hospital, respectively. The patients' $PaO_2/FiO_2$ ratios were assessed at baseline (within 24 h prior to bevacizumab administration), on days 1 and 7; chest CT was performed at baseline (within 48 h prior to bevacizumab administration) and day 7 (±1 day), or alternatively performed chest X-ray at baseline (within 48 h prior to bevacizumab administration) at days 3 and 7; laboratory tests including blood routine and C-reactive protein (CRP) at baseline (within 48 h prior to bevacizumab administration) and day 7 was added into the protocol conducted in the Italian population after trial commencement because we observed a phenomenon of rapid abatement of fever post-bevacizumab administration in the middle of the study. Clinical data including demographic data, presenting symptoms, the changes of status of oxygen-support as well as the symptom of fever were recorded using case record forms. This information was subsequently entered into an electronic database and validated by trial staff. Chest CT images were quantified using a medical imaging diagnosis support platform (Huiying Medical Technology Co. LTD, Beijing) to obtain the volume and the ratios of the lesions in bilateral lungs. X-ray images were semi-quantified by the Committee of Imaging Experts using the above platform to assess the ratios of the lesions in bilateral lungs.

**Statistical analysis**. No sample-size calculations were performed. The population under analysis included eligible patients who received a single dose of bevacizumab, and for whom clinical data were available. Categorical variables were reported as numbers and percentages, and quantitative variables as means and standard deviations or medians and quartiles. To compare differences between different time points after intervention and baseline point, a paired $t$-test or Wilcoxon matched-pairs signed-rank test was used for quantitative data. Statuses of oxygen-support and body temperature during 28 days in the bevacizumab-treated population were described with Gantt charts. Safety measurements applied to all patients exposed to bevacizumab. Statistical significance was indicated by a $p$-value of 0.05 and was determined with the use of a two-sided hypothesis test. Wilcoxon signed-rank test and $\chi^2$ test were used to compare the external control and bevacizumab group. All analyses were conducted with SAS software, version 9.4 (SAS Institute, USA).

**Reporting summary**. Further information on research design is available in the Nature Research Reporting Summary linked to this article.

## Data availability

Source data are provided with this paper. Source data are also available in the figshare.com repository at https://doi.org/10.6084/m9.figshare.13482810.

## Code availability

All relevant codes used in SAS software for statistical analysis are provided in the figshare.com repository at https://doi.org/10.6084/m9.figshare.13482834.

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

## Acknowledgements

We thank all the patients who participated in the trial. We thank all the physicians who risked their lives to save patients. This study was supported by grants from the National Key R& D Program of China (2020YFC0846600) and the Shandong Provincial Key R& D Program (2020SFXGFY03). The laboratory of Y.C. is supported through research grants from the European Research Council (ERC) advanced grant ANGIOFAT (project no 250021), the Swedish Research Council, the Swedish Cancer Foundation, the Strategic Research Areas (SFO)–Stem Cell and Regeneration Medicine Foundation, the Karolinska Institutet, the Swedish Children's Cancer Foundation, the Karolinska Institutet Foundation, the Karolinska Institutet Distinguished Professor Award, the Torsten Soderbergs Foundation, the Maud and Birger Gustavsson Foundation, the Novo Nordisk Foundation–Advance grant, and the Knut and Alice Wallenberg's Foundation. The supporting sources have no involvement in this study.

## Author contributions

J.P., F.X., and G.A. are joint first authors. Y. Cao generated the idea of blocking VEGF and targeting the vasculature for effective treatment of patients with Covid-19. Y. Cao and Y. Chen designed the study. F.X. and M.L. participated in the study design. J.P., G.A., Y.B., B.W., W.S., and Ying Zhang managed or helped in the implementation of the project. G.A., Y.L., A.F., G.V., J.W., Y.B., M.C., G.D., J.C., X.Ji., A.K., M.R., and H.W. screened and recruited participants, collected clinical data and images. D.Y. supervised the analysis of images. Yuan Zhang and S.W. managed the data and performed the statistical analysis. J.P. wrote the first draft of the paper. Y. Cao, Y. Chen, F.X., M.L., and X. Jing helped to revise the paper. All authors approved the final version as submitted to the journal.

## Competing interests

The authors declare no competing interests.

## Ethical statement

We complied with all relevant ethical rules. The protocol and informed consent forms were approved by ethics committees of Qilu Hospital of Shandong University, Renmin Hospital of Wuhan University, and Italia Hospital S.p.A. Ospedale Generale di Zona Moriggia – Pelascini, Gravedona ed Uniti (CO) prior to study implementation.
