## [Peer Review File · Nature Communications]

Reviewers' Comments:

Reviewer #1:

Remarks to the Author:

This manuscript describes the results of a single-arm Phase II trial of Bevacizumab for the treatment of patients suffering severe pulmonary distress as a result of a SARS-CoV-2 infection. The topic is novel and has tremendous importance for public health. My comments will focus on the statistical design, analyses and inference.

This preliminary evaluation of bevacizumab is made through two comparisons: 1) a comparison of post-treatment measures to baseline (pre-treatment) measures; 2) a comparison to untreated controls who were selected to be similar to the trial participants. The results seem encouraging however there are several questions.

How were patients identified for screening and consent for treatment at each site and how were the controls identified/selected? According to p 16, lines 2-4, the authors "screened patients with severe Covid-19 who were eligible as control cases admitted during the same period as the bevacizumab-treated group in two centers, and 26 external control patients were included as a 1:1 ratio to bevacizumab-treated patients." Having concurrent controls from the same centers is very important in this rapidly evolving clinical setting but it is equally important to understand how patients were selected to be treated or not. Since all of the inclusion and exclusion criteria were applied to both groups (except the requirement for consent), how was the trial enrollment conducted and how were controls selected? The authors need to be very explicit in their methodology here to address potential sources of bias in comparing these two groups. Without assurances in these fundamental epidemiologic concepts, the external control group comparisons are not interpretable.

It is not clear why the control group was limited to 26 patients. Was this a pair-wise matched design? If so, the analysis should reflect this.

How was day 1 and day 7 defined for the control group? Was there any attempt to make this comparable to the treated group timeframe for each cohort?

Within the trial itself, the primary outcome of partial arterial oxygen pressure to fraction of inspiration oxygen ratio (PaO₂/FiO₂) was observed to increase by a median of 50.5 on day 1 and 111.0 on day 7. As a statistician and clinical trialist who has not worked in pulmonary disease before, I am not familiar with these measures so the clinical relevance isn't obvious. This increase is smaller than the difference noted between the two sites (China and Italy) at baseline. Does that mean that this level of improvement is modest relative to the effects of other factors? Do the differences noted in Table 1 account for these findings? What about other factors (e.g., BMI, smoking, asthma other supportive care provided)?

The authors indicate that 77% of patients had increased PaO₂/FiO₂ but this could include patients who had negligible improvement. If there is an established threshold for a clinically relevant change in PaO₂/FiO₂, it would be more compelling to describe the fraction of patients who achieved that level of oxygenation.

The clinical trial registration indicates that the primary outcome was to be measured on Days 1, 3 and 7. Why were the data for Day 3 not included? It seem that for such a small trial, the data could be displayed more comprehensively, showing the patterns over time in PaO₂/FiO₂ and possibly other outcomes (e.g., temperature/fever).

The analyses comparing the treated group to the controls does not seem to include any adjustment for potential confounding factors other than center. The small sample size precludes any extensive modeling but it seems important to explore the impact of including other factors in the analysis.

Limitations of this study should also include the non-randomized, uncontrolled nature of this trial, the short term follow-up, the high, unexplained variability in baseline PaO₂/FiO₂ between the centers.

Detailed comments:

p. 13 line 3. The use of the word "assigned" in this context is a bit confusing since all enrolled patients were given the same treatment.

p. 14 line 4: Defined "marked improvement"

p. 14 line 19: Does "significantly" meant to refer to statistical or clinical significance? Throughout, it would be helpful to be explicit in the use of this word.

p. 16 line 7: It would be helpful to describe "standard care". Was it the same in both centers?

p. 17 line 8-9: Sentence structure is a bit confusing. The observation is that 92% of patients on bevacizumab experienced improvement during 28 days—not that there was a 92% improvement.

Table 1 : If the control group is retained, suggest combining Table 5 with Table 1. Add the following variables: disease severity, and body mass index, smoking status, and any other relevant comorbidities (asthma)

Table 2: Suggest replacing with a graphical display showing all datapoints (including those from Table 6, if the control group is retained)

Reviewer #2:

Remarks to the Author:

This study presents an promising therapeutic approach for severe Covid-19 patients, which blocks vascular endothelial growth factor (VEGF). A single-arm trial with an external comparison cohort was performed. I have a few comments:

1. Although the manuscript shows that the trial cohort and the comparison cohort are comparable in terms of baseline clinical and demographic characteristics, it is important to clarify how the external comparison cohort was constructed. Specifically, was it constructed prospectively or retrospectively? How the subjects were selected, out of how large a pool? Were the decision-maker blinded to the outcomes? A prospectively enrolled concurrent comparison cohort would be more convincing.

2. Line 173, "the external comparison cohort ... met the inclusion criteria as well as the exclusion criteria". Please clarify the meaning of "the comparison cohort meeting the exclusion criteria".

3. The comparison cohort does not have to be of the same size as the trial cohort. The authors might consider a larger comparison cohort (with comparable characteristics) to provide a more comprehensive picture under control.

Reviewer #3:

Remarks to the Author:

Main finding of the study :

Jiaojiao P. et al. in their submitted article targeted to present the efficiency of Bevacizumab treating Covid-19 patients by conducting a multicenter single-arm clinical trial. A single dose of Bevacizumab of 7.5mg/kg could make a rapid improvement of PaO₂/FiO₂ values, fever, lymphocyte counts and anti-inflammation according to the result of this study.

Strength and limitation :

This study showed an encouraging efficiency concerning the use of a single dose of anti-VEGF treatment in COVID-19 and it is obviously a very interesting approach. It is a sufficiently established

study with promising conclusions.

There are some limitations regarding limited population size, lack of data and language.

Comments :

As a general comment, authors need to clarify all the abbreviations in the text meaning that this should be done the first time that an ab is presented and also a table with all the abs included.

Considering the result of external controls, it is not clear that how the authors had chosen the day 0 in these patients as they did not receive bevacizumab. It should be noticed if the baseline characteristics and data were collected on admission. The prognosis of external controls was also not clarified (ratio of death).

The use of anticoagulation should be mentioned, as the increase of D-dimer is a typical feature in COVID-19 and anticoagulation has been widely used.

Table 4 showed the list of adverse events of patients in this study, and the author noticed an absence serious adverse events after Bevacizumab in line 399-400. The severity of adverse events should be presented according to CTC. What other concomitant diseases or medication might influence on adverse events in this study?

Lack of information of fever of several patient in Figure 5-A, and the reason was not mentioned.

Conclusion:

Acceptance is recommended after minor revision.

Reviewer #4:

Remarks to the Author:

the author presented a one-arm study together with a external control group to show the efficacy and safety of using VEGF-antibody bevacizumab on P/F ratio and other clinical outcomes of severe COVID-19 patients in China and Italy. The results showed rapid improvement of above clinical outcomes both in a one-arm study (before and after administration of bevacizumab) as well as compared to external control group. Based on the potential mechanism and current data, it is very likely that bevacizumab may block overexpressed VEGF, which is responsible for increased vascular permeability in COVID-19 patients, although this is not a RCT and large clinical trials, the prompt and promising results suggested additional and quick clinical trials are needed given the current situation of COVID-19 pandemic worldwide. The current results highlighted the significance of targeting VEGF as one of the potential therapeutic approach against SARS-CoV-2. Of course, some questions need to be clarified:

1. lab results to show any association with VEGF blocking. At least CRP, IL-6, D-dimer etc may reveal the cytokine storm changes before and after bevacizumab administration.
2. any plasma samples saved for this study? if yes, VEGF level need to be measured.
3. Any other parameters to show the vascular permeability changes before and after bevacizumab administration. Such as ICAM-1, VWF.
4. coagulation and fibrinolysis factors changes?
5. How was the steroid and heparin used?

**Karolinska
Institutet**

Yihai Cao, M.D., Ph.D.
Professor of Vascular Biology
Laboratory of Angiogenesis Research
Department of Microbiology, Tumor and Cell Biology
Stockholm, Sweden
Tel: +46-8-524 87596, Fax: +46-8-524 87462
E-mail: yihai.cao@ki.se

Ref. Manuscript NCOMMS-20-27719

September 29, 2020

We thank all the reviewers for the excellent comments, which are of great help to improve our work. Our point-by-point responses are as follows:

Reviewer #1 (Remarks to the Author):

Comment: This manuscript describes the results of a single-arm Phase II trial of Bevacizumab for the treatment of patients suffering severe pulmonary distress as a result of a SARS-CoV-2 infection. The topic is novel and has tremendous importance for public health. My comments will focus on the statistical design, analyses and inference.

Response: We thank the reviewer for correct understanding our study and considering the topic being novel and tremendous importance for public health. These comments are extremely encouraging.

Comment: This preliminary evaluation of bevacizumab is made through two comparisons: 1) a comparison of post-treatment measures to baseline (pre-treatment) measures; 2) a comparison to untreated controls who were selected to be similar to the trial participants. The results seem encouraging however there are several questions.

Response: Agree. Our clinical assessments of the therapeutic efficacy of bevacizumab were obtained through post-treatment vs. pretreatment and a comparison with non-treated controls.

Comment: How were patients identified for screening and consent for treatment at each site and how were the controls identified/selected? According to p 16, lines 2-4, the authors “screened patients with severe Covid-19 who were eligible as control cases admitted during the same period as the bevacizumab-treated group in two centers, and 26 external control patients were included as a 1:1 ratio to bevacizumab-treated patients.” Having concurrent controls from the same centers is very important in this rapidly evolving clinical setting but it is equally important to understand how patients were selected to be treated or not. Since all of the inclusion and exclusion criteria were applied

to both groups (except the requirement for consent), how was the trial enrollment conducted and how were controls selected? The authors need to be very explicit in their methodology here to address potential sources of bias in comparing these two groups. Without assurances in these fundamental epidemiologic concepts, the external control group comparisons are not interpretable.

Response: There are very valuable comments. We apologize for not being explicit in the methodology section to provide detailed information about selection of controls at each site. In the revised manuscript, we have provided this information, which can be summarized as follows:

- 1) We followed the trial protocol at both China and Italy sites. The timeline of recruiting the treated group in China was between the February 15th to March 8th 2020, and in Italy was between the March 25th to April 5th in Italy. The time difference was created because of the different timeline of Covid-19 outbreaks in these two countries, i.e., later outbreak in Italy. When patients were diagnosed for Covid-19 infection, they were admitted to the designated wards in both centers. The medical specialists screened for severity of Covid-19 patients based on the respiratory rate, the oxygen saturation, and the PaO₂/FiO₂ ratio. For those who suffered from respiratory distress, the oxygen saturation <93%, and the PaO₂/FiO₂ ratio <300 mmHg were defined as severe patients. Using this standard, severe patients were selected on the daily basis in both sites and were informed about the trial purpose, study drug, interventions, laboratory, study visits, benefits, and possible harms. Approved by the patients' consent and signing of the informed consent form by patients or authorized legal representatives and the investigators, the investigators carefully evaluated inclusion and exclusion criteria. For further meeting the recruitment criteria, radiographic examinations including lung X-ray or CT scan were employed to assess pulmonary lesions. On the basis of these criteria, patients were confirmed of inclusion in this study.
- 2) For controls, we conducted retrospective screening of patients with severe Covid-19 who had complete dataset of the PaO₂/FiO₂ in the same center within the similar timeframe (\pm 5 days), i.e., February 10th to March 13th, 2020 in China and March 20th to April 10th, 2020 in Italy. In China, a total number of 24 patients with severe Covid-19 were identified as potential controls. Owing to mismatched timepoints of PaO₂/FiO₂ measurements of 10 patients, 14 patients were eligible to serve as controls. Among these 14 patients, 2 participated other trials in due time and finally 12 patients were used as controls. In Italy, a total number of 17 patients with severe Covid-19 were identified. Of the 17 patients, 2 patients with the PaO₂/FiO₂ ratio of less than 100 mmHg were excluded and 1 patient was excluded because of sudden deterioration and a life-threatening condition. Finally, 14 patients were deemed as eligible as controls. The Ethical committees in both centers approved the observational data collection of patients with Covid-19 for the purpose as controls.

- 3) On page 16, lines 2-4, our statement about 1;1 ratio was misleading. We apologize for this mistake. The 26 controls from both centers as described were coincidentally to be the number as the treated group, which was not intentionally designed.

Prospectively concurrent controls or randomized controls would be desirable to achieve statistical efficacy. However, owing to the panic pandemic situation in Wuhan and Lombardia during February and March of 2020, which were the most heavily affected areas around the globe, investigators in both centers had limited resources, medical devices, and stressful working environment to design a complete prospective trial. Therefore, we had done our best to select controls for our study. We thank the reviewer for his/her understanding. In the revised version of the manuscript, we have now provided the detailed information of identifying controls and the consent for treatments.

Comment: It is not clear why the control group was limited to 26 patients. Was this a pair-wise matched design? If so, the analysis should reflect this.

Response: We apologize for not providing detailed information about the 26 control patients, which has caused confusion for the reviewer. This was not a pair-wise matched design for controls. For controls, we conducted retrospective screening of patients with severe Covid-19 who had complete dataset of the PaO₂/FiO₂ in the same center within the similar time frame (± 5 days), i.e., February 10th to March 13th in China and March 20th to April 10th in Italy. In China, a total number of 24 patients with severe Covid-19 were identified as potential controls. Owing to mismatched timepoints of PaO₂/FiO₂ measurements of 10 patients, 14 patients were eligible to serve as controls. Among these 14 patients, 2 participated other trials in due time and finally 12 patients were used as controls. In Italy, a total number of 17 patients with severe Covid-19 were identified. Of the 17 patients, 2 patients with the PaO₂/FiO₂ ratio of less than 100 mmHg were excluded and 1 patient were excluded because of sudden deterioration and a life-threatening condition. Finally, 14 patients were deemed as eligible as controls. The 26 controls from both centers as described were coincidentally to be the same number as the treated group, which was not intentionally designed. In the revised manuscript, we have now provided the detailed information of selecting patients.

Comment: How was day 1 and day 7 defined for the control group? Was there any attempt to make this comparable to the treated group timeframe for each cohort?

Response: These are excellent questions. The reviewer is completely correct. Day 1 and day 7 in control groups were defined to be comparable to the treated group timeframe for each cohort. The period from the admission time to day 0 was comparable between the external control group and the bevacizumab-treated group in the same center. In the revised manuscript, we have clarified this important point.

Comment: Within the trial itself, the primary outcome of partial arterial oxygen pressure to fraction of inspiration oxygen ratio ($\text{PaO}_2/\text{FiO}_2$) was observed to increase by a median of 50.5 on day 1 and 111.0 on day 7. As a statistician and clinical trialist who has not worked in pulmonary disease before, I am not familiar with these measures so the clinical relevance isn't obvious. This increase is smaller than the difference noted between the two sites (China and Italy) at baseline. Does that mean that this level of improvement is modest relative to the effects of other factors? Do the differences noted in Table 1 account for these findings? What about other factors (e.g., BMI, smoking, asthma other supportive care provided)?

Response: We thank the reviewer for raising these important points. The primary outcome of partial arterial oxygen pressure to fraction of inspiration oxygen ratio ($\text{PaO}_2/\text{FiO}_2$) is a reliable and commonly used parameter for assessing oxygen saturation and pulmonary disease. These are well defined values during clinical practice for severity of respiratory disease: 1) the normal value is within a range 400-500 mmHg; 2) 200-300 as mild acute respiratory distress syndrome (ARDS); 3) 100-200 mmHg as moderate ARDS; and with <100 mmHg as severe ARDS. In our experiences of both Chinese and Italian sites, the median increases of $\text{PaO}_2/\text{FiO}_2$ by 50.5 mmHg and 111.0 mmHg on day 1 and day 7, respectively were considered to very significant, especially on day 1, the quick improvement of hypoxemia was remarkable.

The baseline difference between the Italian and Chinese patients was likely generated by the variation of Covid-19 infection at different stages upon enrollments in their respective hospitals. As shown in Table 1, the time of "admission to BEVA treatment was obviously different between the two sites. The median of interquartile range (IQR) in the Italian group was 4 days (3,6), where the median IQR for the Chinese group was 12 (9, 24). Importantly, comparison of $\text{PaO}_2/\text{FiO}_2$ values within Italian and Chinese groups showed significant improvements (Revised Figure 2). In general, the Italian patients were at the early stage of Covid-19 infection with more severity relative to the Chinese group.

On the basis of the reviewer's recommendation, we have included additional parameters, including BMI, smoking, asthma, supportive care in the revised manuscript (Revised Table 1). We have now included these additional information in the revised version of the manuscript.

Comment: The authors indicate that 77% of patients had increased $\text{PaO}_2/\text{FiO}_2$ but this could include patients who had negligible improvement. If there is an established threshold for a clinically relevant change in $\text{PaO}_2/\text{FiO}_2$, it would be more compelling to describe the fraction of patients who achieved that level of oxygenation.

Response: We thank the reviewer for the valuable suggestion by establishing thresholds of $\text{PaO}_2/\text{FiO}_2$. We have defined a threshold of $\text{PaO}_2/\text{FiO}_2$ increase by 50 mmHg and 100 mmHg from the baseline 100 mmHg (relatively high thresholds) for day

1 and day 7, respectively. On day 7, according to the stratification of ARDS as indicated above, achieving these thresholds means to gain one class from moderate ($\text{PaO}_2/\text{FiO}_2$ 100-200 mmHg) to mild ($\text{PaO}_2/\text{FiO}_2$ 200-300 mmHg), which was an excellent proxy of clinical global prognosis. An increase of $\text{PaO}_2/\text{FiO}_2$ by 100 mmHg likely presents healing and shortness of hospitalization, and most of all, reducing the possibility of intubation and receiving further treatment in ICU. Optimization of resources, limitation of ICU beds, and sustainability are crucial clues for any hospital, especially in dramatic situation such as this one. After establishing thresholds, the results showed that 13 patients (50%) on day 1 and 15 patients (57.7%) on day 7 in the bevacizumab-treated group reached the thresholds whereas only 5 patients (19.2%) on day 1 and 4 patients (15.4) on day 7 in the control group. The net increase on day 1 was by 30,8% and day 7 by 42.3%, which were considered to substantial increases. We have now included these new data in the revised manuscript (Supplementary Figure 1).

Comment: The clinical trial registration indicates that the primary outcome was to be measured on Days 1, 3 and 7. Why were the data for Day 3 not included? It seems that for such a small trial, the data could be displayed more comprehensively, showing the patterns over time in $\text{PaO}_2/\text{FiO}_2$ and possibly other outcomes (e.g., temperature/fever).

Response: We thank the reviewer for this important comment. Originally, we planned to have 3 measurements at Day 1, 3, and 7. However, during execution of the trial protocol in Wuhan, data collection of day 3 were omitted due to the following reasons: 1) This trial was initiated at the peak time of pandemic outbreak both in Wuhan. There were not sufficient medical facilities were available on day 3 for measurements; and 2) The Wuhan site lacked sufficient numbers of medical personals to collect data on day 3. The hospitals at that time were overwhelmed with hospitalized patients. We had realized that it was a tremendously difficult situation at the peak time of Covid-19 outbreak to execute a complete trial protocol. It was an unexpected challenging situation for medical specialists, hospital loads, leaders, and decision makers.

We had the complete dataset of day 3 from the Lombardian site. These $\text{PaO}_2/\text{FiO}_2$ data are now included in the revised version of the manuscript (Revised Fig. 2). As expected, the day 3 data were statistically significant. We were also able to collect data from 28-day follow-up, which are commonly used for clinical study of pulmonary infectious disease. We thank the reviewer for his/her understanding.

Comment: The analyses comparing the treated group to the controls does not seem to include any adjustment for potential confounding factors other than center. The small sample size precludes any extensive modeling but it seems important to explore the impact of including other factors in the analysis.

Response: We agree with the reviewer's comments. We had tried our best to adjust the cofounding factors, including age, gender, hypertension, heart disease, chronic obstructive between the two groups pulmonary disease, and the baseline of the $\text{PaO}_2/\text{FiO}_2$ values to match the treated groups and controls. As the reviewer suggested, we have in

the revised manuscript also included BMI, smoking, and asthma history as additional cofounding factors for adjustment (Revised Table 1).

Comment: Limitations of this study should also include the non-randomized, uncontrolled nature of this trial, the short term follow-up, the high, unexplained variability in baseline PaO₂/FiO₂ between the centers.

Response: Completely agree. We thank the reviewer for these valuable suggestions. In the revised manuscript, we have stated these limitations in the section of Discussion.

Comment: Detailed comments:

p. 13 line 3. The use of the word “assigned” in this context is a bit confusing since all enrolled patients were given the same treatment.

Response: Completely agree. We have replaced the word “assigned” with “received” in the revised manuscript.

Comment: p. 14 line 4: Defined “marked improvement”

Response: Agree. We have deleted the word “marked” in the revised manuscript.

Comment: p. 14 line 19: Does “significantly” meant to refer to statistical or clinical significance? Throughout, it would be helpful to be explicit in the use of this word.

Response: In this sentence, “significantly” meant statistically significant. We have carefully checked significant and significance throughout the manuscript and to ensure this word means statistical significance.

Comment: p. 16 line 7: It would be helpful to describe “standard care”. Was it the same in both centers?

Response: We thank the reviewer for this excellent comment. In the revised manuscript, we have now described the standard care in both sites, explaining the common regimens and differences. The basic regimens for treating patients with severe Covid-19 were similar in the Wuhan and Lombardian sites, which included antiviral drugs, hydroxychloroquine, antibiotics, steroids, antipyretics, and supportive care. However, there were differences between the two sites. At the Lombardian site, all patients received treatment with anticoagulants, whereas only 2 patients at the Wuhan site received anticoagulants. According to the standard care recommended by the Healthy

Ministry of China, Chinese herb medicine were used for treating all patients with severe Covid-19 and Chinese traditional medicine was not used at the Lombardian site.

Comment: p. 17 line 8-9: Sentence structure is a bit confusing. The observation is that 92% of patients on bevacizumab experienced improvement during 28 days—not that there was a 92% improvement.

Response: Completely agree. In the revised manuscript, we have revised this statement by describing 92% of patients...

Comment: Table 1: If the control group is retained, suggest combining Table 5 with Table 1. Add the following variables: disease severity, and body mass index, smoking status, and any other relevant comorbidities (asthma)

Response: We thank the reviewer for this suggestion. On the basis of the reviewer's suggestion, in the revised manuscript we have combined Table 5 with Table 1. We also added other variables including BMI, smoking status, and asthma in the revised Table 1.

Comment: Table 2: Suggest replacing with a graphical display showing all datapoints (including those from Table 6, if the control group is retained)

Response: We thank the reviewer for this suggestion. In the revised manuscript, we have replaced Table 2 with a graphic display with all datapoints (Figure 2).

Reviewer #2 (Remarks to the Author):

Comment: This study presents a promising therapeutic approach for severe Covid-19 patients, which blocks vascular endothelial growth factor (VEGF). A single-arm trial with an external comparison cohort was performed. I have a few comments:

Response: We thank the reviewer for considering our study being promising for treating patients with severe Covid-19.

Comment: 1. Although the manuscript shows that the trial cohort and the comparison cohort are comparable in terms of baseline clinical and demographic characteristics, it is important to clarify how the external comparison cohort was constructed. Specifically, was it constructed prospectively or retrospectively? How the subjects were selected, out of how large a pool? Were the decision-maker blinded to the outcomes? A prospectively enrolled concurrent comparison cohort would be more convincing.

Response: We thank the reviewer for raising these important questions. The external comparison cohort was constructed retrospectively. For controls, we conducted retrospective screening of patients with severe Covid-19 who had complete dataset of the PaO₂/FiO₂ in the same center within the similar time frame (± 5 days), i.e., February 10th to March 13th in China and March 20th to April 10th in Italy. In China, a total number of 24 patients with severe Covid-19 were identified as potential controls. Owing to mismatched timepoints of PaO₂/FiO₂ measurements of 10 patients, 14 patients were eligible to serve as controls. Among these 14 patients, 2 participated other trials in due time and finally 12 patients were used as controls. In Italy, a total number of 17 patients with severe Covid-19 were identified. Of the 17 patients, 2 patients with the PaO₂/FiO₂ ratio of less than 100 mmHg were excluded and 1 patient was excluded because of sudden deterioration and a life-threatening condition. Finally, 14 patients were deemed as eligible as controls. The Ethical committees in both centers approved the observational data collection of patients with Covid-19 for the purpose as controls. The decision makers were not blinded to the outcome. We agree that a prospective concurrent comparison would have been more compelling and convincing. Unfortunately, designing a prospective cohort as a control during the peak time of outbreak in Wuhan and Lombardian was extremely challenging owing to limited resources and ethical regulations. We thank the reviewer for his/her understanding. In the revised manuscript, we have now described the detailed information of selection external controls.

Comment: 2. Line 173, “the external comparison cohort ... met the inclusion criteria as well as the exclusion criteria”. Please clarify the meaning of "the comparison cohort meeting the exclusion criteria".

Response: Agree. We apologize for the confusion regarding the external comparison cohort. We did not accurately describe the exact meaning of the exclusion criteria in the original version of the manuscript. What we meant was the inclusion criteria. In the revised manuscript, we have deleted the words “as well as exclusion criteria”.

Comment: 3. The comparison cohort does not have to be of the same size as the trial cohort. The authors might consider a larger comparison cohort (with comparable characteristics) to provide a more comprehensive picture under control.

Response: Completely agree. The 26 controls from both centers as described were coincidentally to be the same number as the treated group, which was not intentionally designed. In the revised manuscript, we have now provided the detailed information of selecting patients. As responded in Point 1, there were not large pools of concurrent external controls to be selected in both Wuhan and Lombardian sites. For those who did not match timepoints of measurements, baseline PaO₂/FiO₂, participation in other trials, and unexpected life-threatening conditions were excluded. Otherwise they were enrolled as external comparison cohorts.

Reviewer #3 (Remarks to the Author):

Comment: Main finding of the study:

Jiaojiao P. et al. in their submitted article targeted to present the efficiency of Bevacizumab treating Covid-19 patients by conducting a multicenter single-arm clinical trial. A single dose of Bevacizumab of 7.5mg/kg could make a rapid improvement of PaO₂/FiO₂ values, fever, lymphocyte counts and anti-inflammation according to the result of this study.

Response: We thank the reviewer for correctly understanding our findings.

Comment: Strength and limitation:

This study showed an encouraging efficiency concerning the use of a single dose of anti-VEGF treatment in COVID-19 and it is obviously a very interesting approach. It is a sufficiently established study with promising conclusions. There are some limitations regarding limited population size, lack of data and language.

Response: We thank the reviewer for considering our therapeutic approach being interesting and therapeutic efficacy is encouraging. We agree that the cohort size of our trial is rather limited owing to the limited availability of patients with severe Covid-19 during a specific period of outbreak in China and Italy. In the revised manuscript, we have carefully read through the manuscript and spotted out typo and other linguistic errors. We believe that the revised manuscript is significantly improved.

Comment: Comments:

As a general comment, authors need to clarify all the abbreviations in the text meaning that this should to be done the first time that an ab is presented and also a table with all the abs included.

Response: Completely agree. We have carefully read through the manuscript and tables to ensure that all abbreviations are spelled out for their first appearance. Now, these abbreviations are in good order of the revised manuscript.

Comment: Considering the result of external controls, it is not clear that how the authors had chosen the day 0 in these patients as they did not receive bevacizumab. It should be noticed if the baseline characteristics and data were collected on admission. The prognosis of external controls was also not clarified (ratio of death).

Response: Agree. For external controls, we conducted retrospective screening of patients with severe Covid-19 who had complete dataset of the PaO₂/FiO₂ in the same

center within the similar timeframe (± 5 days), i.e., February 10th to March 13th in China and March 20th to April 10th in Italy. In China, a total number of 24 patients with severe Covid-19 were identified as potential controls. Owing to mismatched timepoints of PaO₂/FiO₂ measurements of 10 patients, 14 patients were eligible to serve as controls. Among these 14 patients, 2 participated other trials in due time and finally 12 patients were used as controls. In Italy, a total number of 17 patients with severe Covid-19 were identified. Of the 17 patients, 2 patients with the PaO₂/FiO₂ ratio of less than 100 mmHg were excluded and 1 patient was excluded because of sudden deterioration and a life-threatening condition. Finally, 14 patients were deemed as eligible as controls. The Ethical committees in both centers approved the observational data collection of patients with Covid-19 for the purpose as controls. Regarding the prognosis of external controls, 3 patients died of Covid-19 and 0 patient died in the bevacizumab-treated group (11.5% vs. 0%). In the revised manuscript, we have provided this information of selection of external controls and prognosis (Supplementary Fig. 3).

Comment: The use of anticoagulation should be mentioned, as the increase of D-dimer is a typical feature in COVID-19 and anticoagulation has been widely used.

Response: Completely agree. This is an excellent comment. Concerning the use of anticoagulants, there were differences between the Wuhan site and Lombardian site. The outbreak in Wuhan started earlier than Lombardian and at that time anticoagulant was not recommended in the standard regimen by the Health Ministry of China. Therefore, only 2 patients received anticoagulant treatment. However, in Lombardian all patients received anticoagulant therapy, which was a part of standard care in Italy. In the revised manuscript, we have clarified this point.

Comment: Table 4 showed the list of adverse events of patients in this study, and the author noticed an absence serious adverse events after Bevacizumab in line 399-400. The severity of adverse events should be presented according to CTC. What other concomitant diseases or medication might influence on adverse events in this study?

Response: Completely agree. Based on the reviewer's suggestion, we have revised Table 4 and presented the severity of adverse effects according to CTCAE (revised Table 3). There were several possible factors that might influence on the adverse effects in our study, including 1) other medications; 2) Chinese herb medicine; 3) antiviral drugs; 4) genetic variations; and concomitant diseases. We have described these possibilities in the revised manuscript.

Comment: Lack of information of fever of several patient in Figure 5-A, and the reason was not mentioned.

Response: We thank the reviewer for careful reading our manuscript and finding this missing information. Among the 26-treated patients, 14 had fever at the time prior to

bevacizumab treatment and 12 had no fever. In Figure 5, we only presented the 14 patients with fever and did not describe the 12 patients without fever.

Comment: Conclusion:
Acceptance is recommended after minor revision.

Response: We thank the reviewer for the positive recommendation for accepting our manuscript for publication after minor revision.

Reviewer #4 (Remarks to the Author):

Comment: the author presented a one-arm study together with an external control group to show the efficacy and safety of using VEGF-antibody bevacizumab on P/F ratio and other clinical outcomes of severe COVID-19 patients in China and Italy. The results showed rapid improvement of above clinical outcomes both in a one-arm study (before and after administration of bevacizumab) as well as compared to external control group. Based on the potential mechanism and current data, it is very likely that bevacizumab may block overexpressed VEGF, which is responsible for increased vascular permeability in COVID-19 patients, although this is not a RCT and large clinical trials, the prompt and promising results suggested additional and quick clinical trials are needed given the current situation of COVID-19 pandemic worldwide. The current results highlighted the significance of targeting VEGF as one of the potential therapeutic approach against SARS-CoV-2. Of course, some questions need to be clarified:

Response: We thank the reviewer for considering our findings being important and promising. We appreciate the reviewer's expertise in this field and support the fact that VEGF is a valid target for treating patients with severe Covid-19.

Comment: 1. lab results to show any association with VEGF blocking. At least CRP, IL-6, D-dimer etc may reveal the cytokine storm changes before and after bevacizumab administration.

Response: Agree. These are excellent suggestions. We measured CRP and D-dimer. Based on available data, CRP levels were significantly reduced compared to the baseline ($p = 0.02$) at day 7 after bevacizumab treatment. These data are presented in the revised version of the manuscript (Fig. 6c). However, D-dimer levels remained unchanged prior to and after bevacizumab treatment. Since VEGF is mainly involved in vascular functions, we did not measure inflammatory cytokine IL-6. We have taken the reviewer's advice, measurements of CRP, D-dimer, and Il-6 are now included in a RCT trial protocol for our future studies.

Comment: 2. any plasma samples saved for this study? if yes, VEGF level need to be measured.

Response: Unfortunately, we didn't collect the plasma samples due to the safety regulation of handling Covid-19 tissue samples.

Comment: 3. Any other parameters to show the vascular permeability changes before and after bevacizumab administration. Such as ICAM-1, VWF.

Response: These are very good suggestions. Unfortunately, owing the pandemic and stressful challenging situations, we were unable to obtain relevant tissue samples to study the impact of bevacizumab on vascular permeability.

Comment: 4. coagulation and fibrinolysis factors changes?

Response: Agree. In addition to D-dimer, activated partial thromboplastin time (APTT) and fibrinogen were measured in a fraction of patients before and after bevacizumab treatment. There were no significant changes of these fibrinolysis factors. These data are now included in the revised manuscript (Supplementary Table 1).

Comment: 5. How was the steroid and heparin used?

Response: For steroids, 15 of 26 patients received steroid treatment, 7 Italian patients and 8 Chinese patients. For heparin, all the patients in the Italian group and 2 patients in the Chinese group received enoxaparin treatment. The use of steroid and heparin in control patients were comparable to the treated group at the both sites. In Chinese control patients, 8 patients received steroids and 4 received enoxaparin treatments. In Italian control patients, 5 patients received steroids and 6 received enoxaparin treatments. We have provided this information in the revised manuscript.

Sincerely,

Yihai Cao
Professor
Karolinska Institutet
Email: Yihai.cao@ki.se

Reviewers' Comments:

Reviewer #1:

Remarks to the Author:

I want to thank the authors for their thoughtful responses to the prior reviews. I also appreciate that it must have been very difficult to conduct a clinical trial in the context of the early days of the pandemic. I found their explanations for some of the gaps to be helpful and reasonable.

The one remaining issue for me is the the formation of the external control group. It is difficult to understand, either in the current version of the manuscript or the response to the reviewers, how the individuals who received the experimental treatment were selected, given the statement that the controls were also treated at the same institutions during the same general timeframe. Were the external controls not subject to the same exclusion criteria? This could potentially bias the control group toward more serious cases. There is also a statement that controls had complete data PaO₂/FiO₂ within a similar timeframe (Line 205). Does this mean that the controls had to survive through day 7 to be eligible? Ideally, the control group would be selected based using only the information available on the comparable day 0 using as much of the same inclusion/exclusion criteria as possible. Any exceptions to this should be stated explicitly. It would be helpful if the CONSORT diagram represented this in its entirety.

A few specific comments:

Abstract: The sentence beginning on line 65 on the efficacy compared to the control group should follow the primary results of the intervention group. This is essentially a single arm trial—the external control group is there to provide a helpful perspective on these results but a statistical comparison of the two groups is not the primary outcome.

Line 69 and 70: Please indicate what these statistics are—e.g., mean and 95% confidence intervals or median and intraquartile ranges? And are these comparisons to baseline or to the external group?

Line 256-257. How is deterioration defined—just oxygen support? If this is referring to other problems (organ failure, etc), please explain and show the data.

Figures 3 and 6 provide excellent summaries of these data. It would be helpful to have the similar fever information for the control group in supplemental figures.

Reviewer #2:

Remarks to the Author:

The authors have addressed my concerns adequately. No further concerns.

Reviewer #3:

Remarks to the Author:

I would like to thank the authors for the revision and agree with all modifications. I believe that the authors made a sufficient revision of their manuscript and I endorse the publication.

Reviewer #1 (Remarks to the Author):

Comment: I want to thank the authors for their thoughtful responses to the prior reviews. I also appreciate that it must have been very difficult to conduct a clinical trial in the context of the early days of the pandemic. I found their explanations for some of the gaps to be helpful and reasonable.

Response: We thank the reviewer for these positive comments, which are very encouraging for us to improve our work.

Comment: The one remaining issue for me is the formation of the external control group. It is difficult to understand, either in the current version of the manuscript or the response to the reviewers, how the individuals who received the experimental treatment were selected, given the statement that the controls were also treated at the same institutions during the same general timeframe. Were the external controls not subject to the same exclusion criteria? This could potentially bias the control group toward more serious cases. There is also a statement that controls had complete data PaO₂/FiO₂ within a similar timeframe (Line 205). Does this mean that the controls had to survive through day 7 to be eligible? Ideally, the control group would be selected based using only the information available on the comparable day 0 using as much of the same inclusion/exclusion criteria as possible. Any exceptions to this should be stated explicitly. It would be helpful if the CONSORT diagram represented this in its entirety.

Response: We apologize once again for the confusion about the control group. The inclusion criteria between the bevacizumab and control groups were the same. However, the exclusion criteria between the treated group and control group were slightly different. A fraction of patients who did not sign informed consent to receive bevacizumab treatment and those with other relatively stable disorders that had a minimal impact on the respiratory progression or survival of COVID-19 were recruited as controls. Of note, controls were screened from different wards within the same medical center i.e., different wards within the Wuhan or Lombardian medical centers. The controls had complete PaO₂/FiO₂ data, except 2 Italian cases on day 1 due to the limitation of medical resources. The controls did not survive through day 7 to be eligible.

We agree with the reviewer's opinion about the ideal recruitment criteria in the treated and control groups on day 0. Unfortunately, during the early outbreak of Covid-19 pandemic of February through April, it was almost impossible to apply the exact same criteria to the treated and control groups. Additionally, the controls had to meet the complete measurements of PaO₂/FiO₂ during the equivalent trial period. We also agree with the reviewer to include a new Consort Diagram to explicitly describe the recruitment criteria. This new consort diagram is now included in the revised manuscript (Supplementary Fig. 1). We thank the reviewer for his/her understanding.

Comment: A few specific comments:

Abstract: The sentence beginning on line 65 on the efficacy compared to the control group should follow the primary results of the intervention group. This is essentially a single arm trial—the external control group is there to provide a helpful perspective on these results but a statistical comparison of the two groups is not the primary outcome.

Response: Agree. We have now in the revised manuscript described the control group after the primary results. The reviewer is completely correct. Our trial is a single arm trial, primarily comparing clinical outcomes prior to and after treatment, and the external control provides a helpful perspective on these results.

Comment: Line 69 and 70: Please indicate what these statistics are—e.g., mean and 95% confidence intervals or median and intraquartile ranges? And are these comparisons to baseline or to the external group?

Response: Agree. In the revised manuscript, we have specified the statistic values. We have added median, interquartile range (IQR). These comparisons were to baseline, we have added this information in Lines 68-70.

Comment: Line 256-257. How is deterioration defined—just oxygen support? If this is referring to other problems (organ failure, etc), please explain and show the data.

Response: In our study, the deterioration only refers to oxygen-support status. We have rephrased all “deterioration” as “deterioration of oxygen-support status”.

Comment: Figures 3 and 6 provide excellent summaries of these data. It would be helpful to have the similar fever information for the control group in supplemental figures.

Response: Completely agree. According to the reviewer’s recommendation, we have added a new figure (Supplementary Fig. 5) to summarize the fever data of the control group.

Reviewer #2 (Remarks to the Author):

Comment: The authors have addressed my concerns adequately. No further concerns.

Response: We thank the reviewer for accepting our paper for publication.

Reviewer #3 (Remarks to the Author):

Comment: I would like to thank the authors for the revision and agree with all modifications. I believe that the authors made a sufficient revision of their manuscript and I endorse the publication.

Response: We thank the reviewer for endorsing our work for publication.

Once again, we thank you for your advice for further improving our work. We hope the revised manuscript is acceptable for publication in your journal.

Sincerely,

Yihai Cao
Professor
Karolinska Institutet
Email: Yihai.cao@ki.se

Reviewers' Comments:

Reviewer #1:

Remarks to the Author:

I thank the authors for their positive responses to all of my previous comments. I have no other concerns.